# Neural Correlates of a Trance Process and Alternative States of Consciousness in a Traditional Healer

**DOI:** 10.3390/brainsci11040497

**Published:** 2021-04-14

**Authors:** Rebecca G. Rogerson, Rebecca E. Barnstaple, Joseph FX DeSouza

**Affiliations:** 1Centre for Vision Research and Interdisciplinary Graduate Program, York University, Toronto, ON M3P 1P3, Canada; mymakhosi@gmail.com; 2Centre for Vision Research, Department of Dance and Neuroscience Graduate Diploma Program, York University, Toronto, ON M3P 1P3, Canada; rebecca.barnstaple@gmail.com; 3Centre for Vision Research, Department of Psychology, Biology, Interdisciplinary Graduate Program, Canadian Action and Perception Network-CAPnet, Vision Science to Application-VISTA, York University, Toronto, ON M3P 1P3, Canada

**Keywords:** brain, plasticity, trance processes, learning, auditory (A1), area prostriata, dance

## Abstract

Trance processes are a form of altered states of consciousness (ASC) widely reported across cultures. Entering these states is often linked to auditory stimuli such as singing, chanting, or rhythmic drumming. While scientific research into this phenomenon is relatively nascent, there is emerging interest in investigating the neural correlates of altered states of consciousness such as trance. This report aims to add to this field of ASC through exploring how the perception of an experienced *Sangoma* (traditional South African healer) entering a trance process correlates to blood-oxygen-level-dependent (BOLD) signal modulation with auditory stimuli. Functional Magnetic Resonance Imaging (fMRI) data were analyzed using a General Linear Model comparing music versus no music condition multiplied by the percept of experiencing trance (High or Low). Positive BOLD activation was shown in the auditory cortex in both hemispheres during a trance process. Other brain regions tightly correlated to trance perception were the right parietal, right frontal, and area prostriata (*p* < 0.05, Bonferroni corrected). The orbitofrontal cortex (part of the Default Mode Network) was negatively activated and most correlated with music when trance was high, showing the largest differential between high and low trance perception. This is the first study to directly correlate BOLD signal variations in an expert subject’s percept of trance onset and intensity, providing insight into the neural signature and dynamics of this unique form of ASC. Future studies should examine in greater detail the perception of trance processes in expert subjects, adding other neuroimaging modalities to further investigate how these brain regions are modulated by trance expertise.

## 1. Introduction

Altered states of consciousness (ASC) have been described worldwide, often in association with religious or spiritual practices; until recently, however, ASC were understood within biomedical paradigms primarily as a form of psychopathology or a physiological response to stress [1]. While there is growing interest in experimental investigations of ASC such as those associated with hypnosis, meditation, flow states, or disassociation, the neural correlates of trance are still largely unknown [2,3,4,5]. The inclusion of subjective reporting on conscious experience may assist in modeling systemic organization and supervenient effects potentially impacting lower-level brain processes in ASCs, including neurophysiology [6].

Trance processes vary across ethnicities and socio-cultural practices, as well as within traditions, communities, and even among individuals. Therefore, a single definition and understanding of trance cannot be reduced, naturalized, medicalized, or universalized, but it is best recognized “on its own unique terms” [1]. Shared across many trance processes are “alterations or discontinuity in consciousness, awareness, personality, or other aspects of physical functioning” [7]. Chanting, singing, drumming, dancing, and stereotyped movements often play a role in entering and maintaining trance, and may contribute to experiencing *flow*, an absorptive state in which skills and action are perfectly matched, resulting in a “forgetting of all else” [8].

This study examines the neural correlates for a singular trance experience achieved by an expert in the *Ngoma* traditional healing practices of South Africa. While there are many types of traditional South African healers or Traditional Health Practitioners (THPs), our study chose to focus on a *Sangoma* (diviner) as trance states are used for learning, healing, and ancestral engagement in this tradition. The *thwasa,* or healing student, learns during a community-based apprenticeship process how to “call up the ancestors” through a trance process with the use of internal and external auditory stimuli, entering into respective *flow* states. For this study, music was selected by the subject to induce trance within the confines of the Magnetic Resonance Imaging (MRI) machine, with the goal of examining the conscious experience or *perception* of the trance process. The expert subject has over 20 years of experience with trance processes as a practicing *Sangoma*. 

Our approach to this study is scaffolded on simultaneous research involving groups of experts (break-dancers, ballet dancers, choreographers) who regularly engage in music-driven behaviors, such as associated or learned movements and states. Experts have putative specialized neural circuits for behaviors related to their area of expertise that can be probed for brain–behavior relationships [9] and potentially used in training to reduce neurodegeneration [10,11]. These brain–behavior associations can be further elaborated through a “first-person neuroscience” approach [12], correlating brain changes with the practitioner’s subjective experience. The aim of this study was to examine brain regions showing an interaction with the expert subjects’ percepts, correlating increases in modulation with a high/low perception of trance. Additionally, anatomical [3] and functionally [13] mapped brain regions of interest were used from [3,13] to guide our inquiry into trance perception.

## 2. Materials and Methods

Participant. One experienced practicing *Sangoma* (female, age 42 years, dance experience of +20 years) underwent fMRI scanning. York University’s ethics committee approved the study (e2013-313), and written informed consent was obtained from the participant in accordance with the committee’s guidelines.

Equipment and Scanning Procedure. A 3T Siemens Tim Trio MRI scanner was used to acquire functional and anatomical images using a 32-channel head coil. T2*-weighted echo planar imaging using parallel imaging (GRAPPA) with an acceleration factor of 2X with the following parameters: 32 slices, 56 × 70 matrix, 210 mm × 168 mm FOV, 3 × 3 × 4 mm slice thick, TE = 30 m, flip angle of 90°, volume acquisition time of 2.0 s, was used. There was a total of 240 volumes per scan. Echo-planar images were co-registered with the high-resolution (1 mm^3^) anatomical scan of the subject’s brain taken at the end of the session (spin echo, TR = 1900 m, TE = 2.52 m, flip angle = 9°, 256 × 256 matrix). The subject’s head was restrained with padded cushions to reduce head movements.

The subject wore headphones (MR Confon, Magdeburg, Germany) to listen to the music while in the scanner and was given a choice as to whether to have eyes open or closed for the duration of the scan. The trance-inducing music task employed a blocked design 30 s *OFF* and 60 s *ON*; *ON* states were alternated five times with 30-s periods of rest for a total scan time of 8-min. Task 2 used the same timing structure and was designed to provide an independent functional localization of right foot motor areas; our subject was visually cued with the word “wiggle” at the center of the projected screen in the scanner and trained on this task before entering the scanner. As the trance process under investigation normally involves movement and/or dance (motor circuitry), the purpose of this task is to map the foot network (see more details in [9,13]), providing an independent measure that can be used to probe brain regions for trance perception. These tasks were analyzed using the General Linear Model (GLM) in BrainVoyager QX (Brain Innovation v2.1.1.1542, Maastricht, The Netherlands) using the boxcar function convolved with a double gamma hemodynamic response function and the subject’s perception of trance [0; 0.5; 1]. Following statistical analysis, blood-oxygen-dependent (BOLD) signal data was analyzed in MATLAB (The MathWorks Inc., Natick, MA, U.S.A. Version 8.4.0.150421, 2014b).

Task Procedure. While in the scanner, there were two tasks: (1) a trance-inducing visualization task cued by music and (2) moving the right foot at 1-Hz, which is an independent motor localization task used in our previous work to functionally map motor areas/circuits that may be used in the visualization task [9]. During the trance visualization task, the subject was told to listen to the music (The music stimulus used was a one-minute excerpt from Swaziland Cultural Group performance: 00:54-02:18. 7 September 2017) and attempt to enter into trance as she would in her practice as a *Sangoma.* Immediately following completion of the scan, we asked the subject to rate the success of each of the five 1-min periods of music and to indicate the extent to which a trance state was achieved (HIGH = 1) or not (LOW = 0.5; NONE = 0). A follow-up interview was conducted after the structural scans were completed and the subject left the MRI suite, during which the subject was invited to elaborate on her experience in the scanner. We chose this method over a button press or other methods requiring a motor response, as this would have overlapped with the BOLD signal.

Preprocessing. Functional data were superimposed on anatomical brain images, aligned on the anterior commissure–posterior commissure (AC-PC) line, and transformed into Talairach space. Functional data from each scan was screened for motion artifacts from head movement or magnet artifacts to detect eventual abrupt movements of the head. In addition, we ensured that no obvious motion artifacts were present in the activation maps. 

Statistical Analysis. The subject’s functional data was analyzed using the General Linear Model (GLM) module with a weighting of a perceptual model convolved with the music block compared to the period where there was no music (fixation blocks—light gray periods from Figure 1 BOLD signal). Bonferroni correction was used (*p* < 0.0001) since it is more conservative than false discovery rate (FDR). Data from GLM regions were extracted, averaged, and pair-wise *t*-tests were used to compare BOLD signal differences from all brain regions (Auditory Cortex (left and right hemispheres), Supplementary Motor Area (SMA), Parietal Activations (PPC), visual cortex, area prostriata, anterior cingulate cortex (ACC), and posterior cingulate (PCC), motor cortex area of the right foot, caudate, and putamen from Basal ganglia. The last three brain areas were derived from Bar and DeSouza (2016) functional locations [13].

## 3. Results

The subject rated the final two out of five 1-min blocks as (HIGH) trance state during her self-selected music. This rate of achieving trance in the confines of the MRI scanner was expected, as the subject was required to remain still and could only visualize dance or other movement behaviors typically used in trance processes. Given the choice to keep eyes open during the scan, the subject did so initially; however, she afterwards reported that due to the sound and disruption of the fMRI machine, she closed her eyes during the bulk of the scanning period. In a follow-up interview, the subject further reported feeling physical sensations of tingling fingers, nausea, and a hyper-awareness to light, odor, and sound normally associated with entering trance, and she saw herself flying or floating above an unknown locale. In her experience, the music invoked an image in which she saw herself dancing with other *Zangoma*, and she felt both intensely present in her body and also outside of it, “not floating or disassociated, but more of a hyper presence in my body with a simultaneous distance from it.” This experience correlated with the highest or strongest point of the trance. The subject also reported rapid fluttering of the eyelids, increased breathing, and an ecstatic feeling or sensation.

The GLM showed brain regions within auditory cortex, area prostriata, visual, parietal cortex, and orbitofrontal (*p* < 0.05, Bonferroni corrected; threshold of k = 8 voxels).

Both hemispheres in the auditory cortex within the temporal lobe showed an increase in BOLD signals when perceived trance was high; Figure 1A (yellow box) and 1B highlight right hemisphere auditory cortex and the BOLD signal from the fMRI scan. The BOLD signal (white line Figure 1B) starts out low (light gray highlighted blocks of time); then, it increases in amplitude soon after the music begins (darker blue); this lasts for a 1-min time period, after which the BOLD signal reduces when there is no music. This pattern is repeated four more times. Brighter blue depicts the blocks of music when the subject perceived a higher perception of trance, as reported in the immediate period following the scan. Blocks were rated by the subject as either HIGH or LOW depending on her subjective experience of having achieved trance through visualization of the process accompanied by somatic sensations or imagery (Rating HIGH = 1; LOW = 0.5; NONE = 0). When we average across perceptual blocks of trance from this auditory region and then parse the subject’s “low” or “high” perception of trance, we observe that there is a significantly higher BOLD signal for HIGH perception versus LOW perception (Figure 2).

The exact same auditory stimulus was played for each 1-min block. When we average across the blocks of 1-min of music for the dark blue and brighter blue blocked regions in Figure 1B, we produce Figure 2A with blue and gray lines for periods with music and white lines for periods with no music. The bar graphs in Figure 2B show times at which the perception of trance was significantly higher than when the same music was perceived as a low trance (paired *t*-test, *p* < 0.05); thus, it was the subject’s *perception* of trance that putatively modulated the auditory cortical signals since the music input to the headphones was identical for all five blocks.

The inset bar graphs in Figure 3A are shown in relation to an idealized model BOLD signal that would be observed for a brain region significantly modulated by the perception of trance. Plotting the ratio of high to low trance perception [*P*], the orange diagonal line in the figure represents where trance signals would be equivalent (*P[high]* = *P[low]*) for a specific brain region. Activity in three brain regions—early right hemisphere visual cortex, left parietal, and cerebellum—fall near this diagonal line and are thus *NOT* significantly modulated by perception of trance (all Ps > 0.05). All other data points fall to the right or below this diagonal orange line, indicating a stronger signal for HIGH perception of trance versus LOW (*p* < 0.05). If these were to the left of the diagonal line, that would have indicated a higher signal for LOW perception of trance (*P[low]*) versus HIGH (*P[high]*). In our results, no functionally mapped brain region was significantly to the left of the line; the right visual cortex data point is slightly to the left, but it is not significantly far enough to fit this alternative model.

We plotted the auditory cortex signal from Figure 1A and Figure 2B as data point B in Figure 3 (yellow data plot and box), which is below the diagonal line, illustrating that there is a significant difference between HIGH and LOW perception of trance (*p* < 0.05). The error bars of this Figure 3B data point plot the standard error of the mean signal across high and low perceptual states, and these error bars do not touch the orange diagonal line.

Additionally, the left auditory cortex also showed an increased BOLD signal for trance perception (green box, Figure 3C), which was further from the line than that from the right hemisphere. This region in right auditory cortex was modulated by the perception of trance (yellow dot and yellow box in Figure 3B), as it was significantly below the line of equality (orange line). For comparison, all other brain regions from the GLM analysis that passed our Bonferroni threshold and spatial voxel clustering [14] are also plotted in Figure 3. All brain regions were mapped with the GLM from anatomical locations were used from relevant studies of [3] and [13]. Area prostriata, right parietal, foot motor area^Bar+DeSouza^, caudate^Bar+DeSouza^, putamen^Bar+DeSouza^, and supplementary motor cortex^Bar+DeSouza^ also showed significantly enhanced signal for high trance perception (i.e., *P[high]* > *P[low]*), with the exception of right visual cortex, left parietal, and cerebellum, whose data points touched the line of unity (i.e., *P[high]* = *P[low]*). We used anatomical locations of PPC^Hove^ and ACC^Hove^ from [3] to further probe our data (red drop shaded area in Figure 3) and found that these areas were also associated with a higher percept for trance (*p* < 0.05; i.e., *P[high]* > *P[low]*). 

Most interestingly, the orbitofrontal cortex (OFC), an area activated in the Default Mode Network (DMN) when a task is stopped and the subject is at rest, was most negatively correlated to the perception of trance and showed the largest difference of high (*p[high]*) compared to low (*p[low]*) trance perception bilaterally (plotted as Figure 3D). OFC was the furthest data point from the diagonal orange line or unity between *p[high]* and *p[low]*, indicating that it was the brain area that showed the largest difference for trance perception in our subject (*p[high]* >> *p[low]*). When we examine the BOLD signals from OFC, there is an expected modulation in the DMN up to the point in which the subject’s perception of trance increased to *high*, at which point the DMN showed decreased activation for the remaining period of data collection (see Figure A1). To our knowledge, this could be the BOLD signal neural correlate of the onset of trance (ASC), as the DMN is turned off, and this OFC BOLD signal does not modulate down when music is back on. This inverse correlation between DMN activity and trance intensity may be due to reciprocal inhibition; DMN and prefrontal areas help maintain Ordinary States of Consciousness (OSCs), and their inhibition correlates with a wide variety of ASCs, including shamanic trance. Similar “inhibition of inhibition” processes have been reported in psilocybin-induced OSCs [15] and in [16] functional hypofrontality studies.

## 4. Discussion

Trance processes represent an intriguing form of ASC with aspects of volitional control not unlike meditation and hypnosis [3,5]. Of particular interest for neuroscience is the correlation of the associated perception of the altered state by an experienced *Sangoma* using self-selected music to induce the trance state. Research into this phenomenon may be a rich source of information as to how attention, volition, activation, and/or suppression of neural networks can play a role in consciousness [17,18,19]. Using trance-inducing music, Hove et al. (2016) found that the effects of drumming on sensory input created patterns of network engagement that they suggest are involved in trance. This included areas in posterior cingulate cortex (PCC), dorsal anterior cingulate cortex (dACC), and left insula/operculum [3]. They also found increased coactivation of the PCC with the dACC and insula, key hubs in the DMN, and executive control networks. Decoupling of seeds within the auditory areas might indicate suppression of the auditory stimuli during trance, with this suppression seeming to play a key role in entering an ASC [3]. Through seed-based functional connectivity and a second seed-based analysis, they found a higher degree of functional connectivity from PPC to ACC/insula, along with clustered activity in the caudal pons, while larger-scale network connectivity was also enhanced during trance. Their seed regions in the auditory cortex suggest it is less connected to PPC and ACC/insula regions. They concluded that trance involves more sustained task maintenance and cooperation of brain networks associated with internal thoughts and cognitive controls, coactive defaults, control networks, and decoupled sensory processing than once believed [3,20].

Using regions from [3] as anatomical brain localizers to probe whether these brain regions are modulated by trance, we found that activity in PPC^Hove^ and ACC^Hove^ was indeed modulated by HIGH perception of trance, but the functionally mapped regions in auditory cortex and OFC regions showed much larger modulation. In a departure from their study [3], we did not rely on data modeling to infer higher trance-related signals in the seed regions; we took a different approach by asking the subject when trance perception was high and low and relied on her subjective experience as a model to run our analysis. While [3] included self-reports following the trance condition to probe whether subjects felt they had entered a trance process, they did not include more nuanced measures of alterations in the degree of trance during this condition. We believe probing the subject’s perceptual shifts during an enhanced trance process adds to their findings and should be part of standard operating procedures to model future trance processes in examinations of ASC studies of trance perception. Our experiment is the first to show increased BOLD signals correlated with the participant’s perception of trance in brain regions activated through music to induce a trance process in an experienced *Sangoma.*

While there has been limited neuroscientific research specifically on trance processes, there is growing interest in the investigation of other kinds of ASC that may suggest future directions for inquiry [21]. The Stanford University School of Medicine conducted a hypnosis-focused study in 2016 with 57 participants, and the research team found decreased activity in the dorsal anterior cingulate and increased activity in the dorsolateral prefrontal cortex and the insula occurred during a hypnotic state. Blanke’s work on Out of Body Experiences (OBE), which he purports are experienced by 2–5% of the population, shows that brain function disruptions also occur for people who experience OBE [22]. A splitting or “body as the observer” mode is experienced, which may be the result of enhanced activity in our right parietal activation near the angular gyrus centered at Talairach coordinates (x = 37; y = −68; z = 25) [22]. Co-activation may also play a role with fluctuations in levels of consciousness, awareness, and sensory patterning (or lack thereof) occurring [20,23,24,25]. The right parietal region also corresponds to studies in [25,26], including [27,28] showing increased metabolic activity in the right parietal regions associated with sensed presences during periods of prayer, thought, and meditation. Our subject’s right parietal brain regions were modulated by HIGH and LOW trance, while the left hemisphere regions were not. Thus, the right parietal region may be important for future studies investigating ASC. Additionally, it should be noted that modulation of activity in frontal polar cortex (BA10), observed during HIGH trance state in our subject, could be a crucial marker of ASC. This region has recently been found to contain von Economo neurons (VENs), associated with social interactions, intuition, and emotional processing [29]. These spindle-shaped neurons, unique to humans, great apes, elephants and cetaceans, have been of particular interest to researchers in consciousness for their proposed role in self-awareness and social cognition. Decreased activity in the DMN during trance for our subject may correlate with differences in activation for VENs in BA10, suggesting a new path for probing the role of these atypical neurons in the production, expansion, or loss of self-awareness in conscious states. 

In Csikszentmihalyi’s Flow: The Psychology of Optimal Experience, he argues that regular “practicing” of, and engagement in, absorptive activities may result in a flow state, which is defined as an automatic, effortless, and highly focused state of mind that occurs when self-reflexive consciousness is harmoniously ordered (2009). Through optimal experience, a sense of exhilaration and deep enjoyment occurs, creating a landmark in the memory for what life should look like [20,30]. The association of trance processes with music suggests the involvement of auditory cortex and areas previously described in a group analysis [3], but what has not been previously described or investigated are brain regions associated with self-referential memories and experiences that may then trigger trance processes. There is a need for studies examining regions involved in memory and emotion coding to investigate whether these may suppress or encourage trance processes. With regard to the latter statement, the area prostriata brain region is situated near input areas of visual and auditory cortex (Rogerson et al., 2018 IMRF abstract); more importantly, it has strong interconnections with the limbic and retrosplenial cortex in anatomical tracing studies [31,32,33,34,35]. The area prostriata is anterior to the visual cortex and has been described as a visual area that acts as an interface for peripheral vision and fast processing [36], and it may also be involved in sending information to multisensory and higher order association areas in the temporal, cingulate, and orbitofrontal cortex [37,38]. Our study is the first to uncover this putative functional connection, as we did not use visual stimuli to functionally activate area prostriata, but rather music associated with the subject’s ongoing trance processes which may or may not include visualization. Auditory drumming has previously been shown to induce signals in the occipital cortex; [39] showed recruitment of the occipital lobe in subjects who had their eyes closed. As this was an EEG study, he could not postulate the source of the signals. In our study, we hypothesize that our subject’s “a priori” experiences with trance (memory areas) may be involved in activating area prostriata when music stimuli are present associated with trance.

## 5. Conclusions

We have aimed to contribute to a reliable neural marker of trance processes [25] in a case study of an experienced *Sangoma.* We have demonstrated an effect of a heightened trance process with associated brain regions that were modulated by perception of trance. This perceived correlation was within the “ON-music-state” bilaterally in auditory cortexes, visual/parietal areas, and most interestingly, modulated over into the default “OFF-state” only within bilateral orbitofrontal cortex. As this is a single case study, we were very conservative in describing the brain regions that were activated, using the most conservative threshold of the Bonferroni multiple comparisons and also used a next neighbor threshold that increased our confidence and reduced our false positive chances. Clearly, there are limitations in this study as to the extent to which we can interpret and extrapolate from a single subject; however, the data explored here are suggestive enough to provide a basis for follow-up study designs. Our future research aims to examine in greater detail trance percepts in this subject and in other similarly skilled subjects. We feel that these kinds of interdisciplinary and collaborative research endeavors may be beneficial to establishing new kinds of clinical approaches and ongoing research projects [8].

## Figures and Tables

**Figure 1 brainsci-11-00497-f001:**
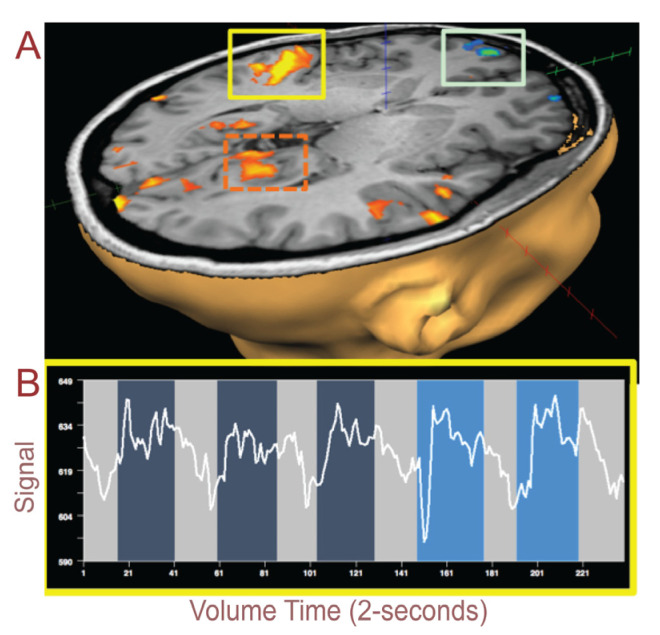
(**A**) Auditory cortex (yellow box) shows an increase in blood-oxygen-level-dependent (BOLD) signals while listening to the music to induce a trance state compared to no music. The light green box surrounds the orbitofrontal activation. The orange dashed box highlights the area prostriata in the functionally defined region. (**B**) BOLD signals extracted from the auditory cortex (yellow box above) show increased signals during music with the last two blocks showing an increase in trance perception, i.e., *P[high]*.

**Figure 2 brainsci-11-00497-f002:**
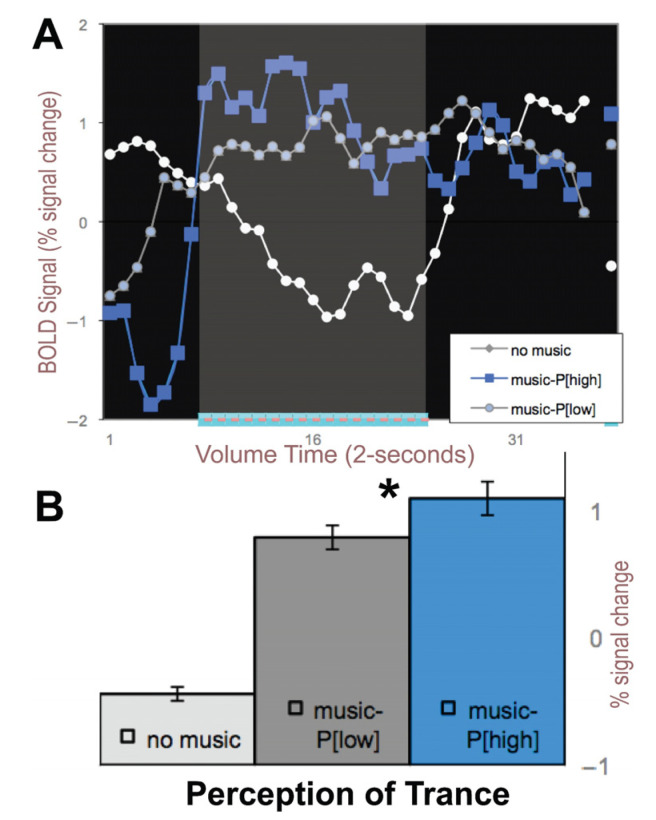
Right auditory cortex BOLD signals averaged across perceptual states of HIGH (*P[high]*) and LOW trance perception (*P[low]*) from Figure 1. (**A**) BOLD signal averaged across the five blocks of 1-min of music and color-coded (blue = high; gray = low) perception of trance. White line depicts when no music was played. (**B**) Average BOLD signal for these states. *- signifies *p* < 0.01 paired *t*-test.

**Figure 3 brainsci-11-00497-f003:**
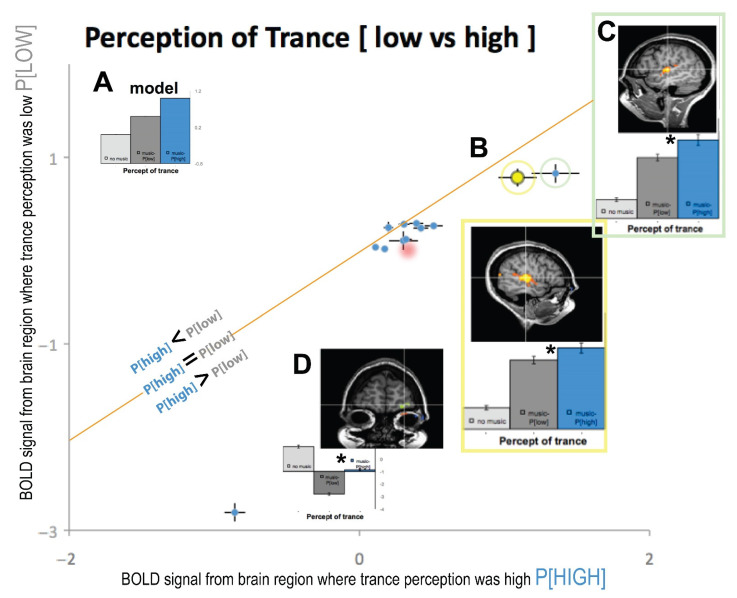
(**A**) The Model BOLD signal if trance has a *P*[*HIGH*] modulated by the music signal. The diagonal orange line represents where low and high trance perception is equal with data below and above the line as the differences. (**B**) Same brain area as in Figure 1A from the right auditory cortex. (**C**) Left auditory cortex bar plots and location. (**D**) Orbitofrontal cortex bar plot and spatial location, Talairach coordinates; x = −24; y = 63; z = 5. Brodmann Area 10. Other data points that were examined from functional ROIs clustered near the orange line. Red drop shading for two data points signifies anatomical regions from [3]. *- denotes significant *p* < 0.05.

## Data Availability

Data is available upon request.

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
