# Peer review of "Neural Correlates of a Trance Process and Alternative States of Consciousness in a Traditional Healer"

_brainsci, 2021, doi:10.3390/brainsci11040497_

Round 1

Reviewer 1 Report

Peer Review

for

Neural Correlates of a Trance Process and Alternative States of Consciousness in a Traditional Healer, Rebecca G. Rogerson, Rebecca E Barnstaple and Joseph FX DeSouza

By

Yakov Shapiro, M.D., F.R.C.P(C)

The authors are addressing an important area of study at the interface of consciousness and neuroscience research. Their findings are generally consistent with the available data on alternate/trance states but there are also novel findings that deserve publication and further research.  However, several methodological and study-specific issues need to be addressed forst, as outlined below.

Introduction:

  1. The introduction is unfocused in several regards. The authors’ focus on perception/action dimension (such as musical cues in dancing) is distinct from the study of ASCs/trance states, which are defined by subjective contents of the associated state of consciousness rather than their motor/behavioral expressions. Therefore, the details of the authors’ previous studies focusing on “specific movement/performance behaviors” (lines 57-65) are more relevant to absorption/flow rather than shamanic/trance states. This methodological issue needs to be addressed.

While drumming, dancing and stereotyped movements are instrumental in many shamanic practitioners, auditory stimulation and movements themselves do not define Shamanic States of Consciousness but rather serve to induce and maintain them in the service of a healing task or shamanic “journey” for the purpose of obtaining task-specific information. Therefore, the study of “brain-behavior relationships” has to be complemented by information about the associated contents of consciousness in these states and the subject’s capacity to use them for a purposeful goal, such as treat “discord” in their tribal members. This requires a “first-person neuroscience” approach (Northoff & Henzel, 2006), correlating brain changes not just with observable movements/behaviors but with the practitioner’s subjective experience.

  1. I would suggest using the term “correlates” rather than “neurological/biological origins” of trance states in line 38. “Origins” suggests a reductive neuroscience bias, where subjective experience is pre-determined by the neurological circuitry and potentially reducible to it. A meta-reductive approach would suggest that the emergent dynamics of conscious experience, including ASCs, has its own level-specific systemic organization and supervenient effects capable of impacting lower-level processes, such as brain neurophysiology (Shapiro & Scott, 2018).

Material and Methods:

  1. The purpose of task 2 (moving the right foot at 1 Hz) is left unexplained. Was task 2 simultaneous with the “ON” condition or carried out throughout the scanning?
  2. The retrospective rating of 5 one-minute sequences in terms of trance intensity (line 99) would pre-suppose accurate recall of all sequences in spite of several transitions between ordinary and trance states of consciousness. How was such accuracy ascertained?
  3. It would appear from the Results that there was a single fMRI trial of 5 musical sequences. Is this in fact how the study was carried out or was there more that one recording session?
  4. What is the function of the follow-up interview (line 100)? Was any subjective information about the content of trance-specific states obtained? If so, it may be helpful to include it in the Results/Discussion.
  5. The choice of the last 3 brain regions (motor cortex of the right foot, caudate and putamen) in lines 114-116 is presumably related to the authors’ previous studies of flow states and task 2 in this study, but their significance for shamanic trance states remains unclear. This needs to be addressed.

Results:

  1. Lines 120-121: “The subject rated two blocks achieving the trance state (High) during her self-selected 120 ” Which 2 blocks out of 5? Figure info would suggest that the last 2 were the highest
  2. Lines 122-123: The sentence “The GLM contrasting perceived trance induced by preferred music was contrasted to fixation showed brain regions…” needs to be clarified
  3. Lines 127-128: “Fig 1A (yellow box) & 1B highlight one hemisphere auditory cortex…” Which hemisphere?
  4. Lines 132-133: “Brighter blue depicts the blocks of music when the subject perceived a higher perception of trance (rated as 1).” Please specify the complete rating system used.
  5. Lines 135-136: “…a significantly higher BOLD signal than for low perception (Fig 2).” This appears to contradict subheading in Fig. 2 indicating higher BOLD for HIGH trance perception
  6. Lines 171-172: “foot motor area Bar+DeSouza, caudate Bar+DeSouza, putamen Bar+DeSouza and supplementary motor cortex Bar+DeSouza also showed stronger activation (Fig 3)” What was the lateralization of these findings? To what extent could these activations be the artefact of task 2 (moving the right foot) rather than shamanic trance per se?
  7. Lines 198-200: “…the orbitofrontal cortex (OFC), an area activated in the default 198 mode network (DMN) when a task is stopped and the subject is at rest, was most negatively correlated to the perception of trance” Please specify right, left, or bilaterally
  8. Lines 209-212: The most likely explanation for the inverse correlation between DMN activity and trance intensity is reciprocal inhibition. DMN and prefrontal areas help maintain Ordinary States of Consciousness (OSCs) and their inhibition correlates with a wide variety of ASCs, including shamanic trance. Similar “inhibition of inhibition” process was observed in psilocybin-induced OSCs (Carhart-Harris et al, 2012) and in Dietrich’s (2003) functional hypofrontality studies.

Discussion:

  1. Lines 217-220: The sentence “Of particular interest for neuroscience is the correlation of the associated experience or perception of the altered state with brain imaging data with our work the first examining perceptual block task related fMRI thus we have greater spatial resolution than.” – needs to be revised
  2. Lines 222-225: The significance of these references is unclear as no specific findings are discussed. In addition, reference 16 is not available on the website provided (it refers to hypnotic rather than shamanic trance). Suggest omitting or discussing further and providing appropriate links.
  3. Lines 246-247: “…we took a different approach by asking the subject when trance perception was high and low and relied on behavior as a model to run our analysis.” It remains unclear what behavior was relied on – please specify.
  4. Lines 261-265. Blanke’s research needs to be referenced directly other than in one of the author’s thesis (ref. 17) if the relevant brain region data is to be used for discussion.
  5. Line 267: Are the authors able to provide more specific localization than a general “right parietal region”? One useful reference source to correlate would be Michael Persinger’s data from Laurentian University (2010).
  6. Lines 291-292: “we did not use visual stimuli to functionally activate area prostriata, but rather music associated with the subject’s ongoing 292 trance processes.” It would be important to specify in the Methods section whether the subject was asked to keep her eyes open or closed during the scan.

Conclusion:

  1. A statement on the limitations of this study should be made.

References:

  1. Northoff, G & Heinzel, A. (2006). First-person neuroscience: A new methodological approach for linking mental and neuronal states. Philosophy, Ethics, and Humanities in Medicine, 1:3
  2. Shapiro, Y & Scott, JR (2018). Dynamical Systems Therapy (DST): Complex Adaptive Systems in psychiatry and psychotherapy. Chapter in: Handbook of Research Methods in Complexity Science: Theory & Application. Prof. E. Mitleton-Kelly, Prof. A. Paraskevas, C. Day (Eds), London: Edward Elgar Publishing LTD
  3. Carhart-Harris, R. L., Erritzoe, D., Williams, T., Stone, J. M., Reed,L. J., Colasanti, A., … Hobden, P. (2012). Neural correlates of the psychedelic state as determined by fMRI studies with psilocybin. Proceedings of the National Academy of Sciences, 109, 2138–2143.
  4. Dietrich A. (2003). Functional neuroanatomy of altered states of consciousness: the transient hypofrontality hypothesis. Consciousness and Cognition, 12(2): 231–256.
  5. Persinger, M. A., Saroka, K. S., Koren, S. A. & St-Pierre, L.S. (2010). The electromagnetic induction of Mystical and Altered States within the Laboratory. Journal of Consciousness Exploration & Research, 1(7): 808-830

Author Response

Thank you for your excellent review of our study.  Please see detailed responses to your comments.

Thank you for your excellent review of our work.  We have made changes and additions throughout the manuscript in response to your suggestions, and feel it is much strengthened.  The new MS has also been reviewed in detail by an editor for English language.

Please see detailed responses to your comments below:

Yakov Shapiro, M.D., F.R.C.P(C)

The authors are addressing an important area of study at the interface of consciousness and neuroscience research. Their findings are generally consistent with the available data on alternate/trance states but there are also novel findings that deserve publication and further research.  However, several methodological and study-specific issues need to be addressed forst, as outlined below.

Introduction:

  1. The introduction is unfocused in several regards. The authors’ focus on perception/action dimension (such as musical cues in dancing) is distinct from the study of ASCs/trance states, which are defined by subjective contents of the associated state of consciousness rather than their motor/behavioral expressions. Therefore, the details of the authors’ previous studies focusing on “specific movement/performance behaviors” (lines 57-65) are more relevant to absorption/flow rather than shamanic/trance states. This methodological issue needs to be addressed.

While drumming, dancing and stereotyped movements are instrumental in many shamanic practitioners, auditory stimulation and movements themselves do not define Shamanic States of Consciousness but rather serve to induce and maintain them in the service of a healing task or shamanic “journey” for the purpose of obtaining task-specific information. Therefore, the study of “brain-behavior relationships” has to be complemented by information about the associated contents of consciousness in these states and the subject’s capacity to use them for a purposeful goal, such as treat “discord” in their tribal members. This requires a “first-person neuroscience” approach (Northoff & Henzel, 2006), correlating brain changes not just with observable movements/behaviors but with the practitioner’s subjective experience.

We have modified the introduction to address the issues above, with a particular focus on clarifying the relationship between the aims and methods of the current study and our previous work. Notably, we’ve specified that the trance process under investigation normally involves dance and movement; thus, while “behaviour” in our study is constrained to subjective reporting (as it involved visualization of behaviour that could not be carried out in the scanner), the phenomena under investigation has many characteristics in common with our previous work. We’ve added details pertaining to the specific trance practice in which the subject is expert, as well as further details on her subjective experience in the Discussion.

  1. I would suggest using the term “correlates” rather than “neurological/biological origins” of trance states in line 38. “Origins” suggests a reductive neuroscience bias, where subjective experience is pre-determined by the neurological circuitry and potentially reducible to it. A meta-reductive approach would suggest that the emergent dynamics of conscious experience, including ASCs, has its own level-specific systemic organization and supervenient effects capable of impacting lower-level processes, such as brain neurophysiology (Shapiro & Scott, 2018).

We have modified the text to “neural correlates of trance” and added details as to how the inclusion of subjective reporting can contribute to better modelling of complex dynamics.

Material and Methods:

  1. The purpose of task 2 (moving the right foot at 1 Hz) is left unexplained. Was task 2 simultaneous with the “ON” condition or carried out throughout the scanning?

Thank you for noticing this.  Now added to line 93: “Task 2 used the same timing structure and was designed to be an independent functional localizer of right foot motor areas; the subject was visually cued with the word “wiggle” at the centre of the projected screen in the scanner, and trained on this task before entering the scanner. As the trance process under investigation normally involves movement or dance (motor circuitry), the purpose of this task is to map the foot network (see more details in Olshansky et al 2014; Bar & DeSouza, 2016), providing an independent measure that can be used to probe for trance perception.”  

  1. The retrospective rating of 5 one-minute sequences in terms of trance intensity (line 99) would pre-suppose accurate recall of all sequences in spite of several transitions between ordinary and trance states of consciousness. How was such accuracy ascertained?

We instructed the subject on this structure before entering the scanner.  We asked the subject immediately to report their perception immediately following the scan, while still in the scanner, and then to elaborate after the scanning session was completed in the interview.  We chose against using a button response during scanning since the motor response of pushing a button would create a dual task with trance perception and motor response overlapped in the BOLD signal, requiring very detailed computations to separate the two overlapping signals.

  1. It would appear from the Results that there was a single fMRI trial of 5 musical sequences. Is this in fact how the study was carried out or was there more that one recording session?

YES - ONE SCAN of 8-minutes long with 5-minutes of 1-minute of music and six 30-second periods of rest which is the exact same protocol we used in previous studies (Olshansky et al 2014; Bar & DeSouza, 2016), the first of which was also a case study.  Although this 8-min scan was short, the power of the auditory stimuli paired with the subject's experience was easily able to push the data to significance even at the most conservative Bonferroni corrections.

  1. What is the function of the follow-up interview (line 100)? Was any subjective information about the content of trance-specific states obtained? If so, it may be helpful to include it in the Results/Discussion.

Now added to lines 142-150. 

  1. The choice of the last 3 brain regions (motor cortex of the right foot, caudate and putamen) in lines 114-116 is presumably related to the authors’ previous studies of flow states and task 2 in this study, but their significance for shamanic trance states remains unclear. This needs to be addressed.

Trance processes share with dance auditory-yoked behaviour that may be modulated by experience/expertise. Our inclusion of motor cortex for right foot has now been explained in the methods, as this is used as an independent localizer in all of our studies where motor behaviour associated with expertise is converted to visualisation due to constraints of the scanner. Caudate and Putamen were activated in the GLM from our previous study Bar & DeSouza 2016.

Results:

  1. Lines 120-121: “The subject rated two blocks achieving the trance state (High) during her self-selected 120 ” Which 2 blocks out of 5? Figure info would suggest that the last 2 were the highest.  This is correct – the last two blocks are highest; this has been clarified in the text.
  2. Lines 122-123: The sentence “The GLM contrasting perceived trance induced by preferred music was contrasted to fixation showed brain regions…” needs to be clarified. This has been addressed.
  3. Lines 127-128: “Fig 1A (yellow box) & 1B highlight one hemisphere auditory cortex…” Which hemisphere?  Right hemisphere; this is now indicated in the text.

Lines 132-133: “Brighter blue depicts the blocks of music when the subject perceived a higher perception of trance (rated as 1).” Please specify the complete rating system used.  Added “Blocks were rated by the subject as either HIGH or LOW depending on her subjective experience of having achieved trance through visualization of the process accompanied by somatic sensations or imagery.”

Also added detail line 163 [Rating HIGH = 1; LOW = 0.5; NONE = 0]

  1. Lines 135-136: “…a significantly higher BOLD signal than for low perception (Fig 2).” This appears to contradict subheading in Fig. 2 indicating higher BOLD for HIGH trance perception.  This has now been corrected on line 165 (unmarked manuscript) by adding “for high perception” and simplification of Figure 2 caption for clarity.
  2. Lines 171-172: “foot motor area Bar+DeSouza, caudate Bar+DeSouza, putamen Bar+DeSouza and supplementary motor cortex Bar+DeSouza also showed stronger activation (Fig 3)” What was the lateralization of these findings? To what extent could these activations be the artefact of task 2 (moving the right foot) rather than shamanic trance per se?  These activations were lateralized due to moving only the right foot in the fMRI scanner and not indicative of the shamanic trance state being lateralized.  These lateralized regions were used as a functional mapping region to sample trance scans in an independent localization of function, in which the subject was visualizing movement.  If we had used the left foot, trance differences would still be evident through the activation windows since the trance state will not be lateralized in the brain.
  3. Lines 198-200: “…the orbitofrontal cortex (OFC), an area activated in the default 198 mode network (DMN) when a task is stopped and the subject is at rest, was most negatively correlated to the perception of trance” Please specify right, left, or bilaterally.  Bilateral; we’ve added this to the text.
  4. Lines 209-212: The most likely explanation for the inverse correlation between DMN activity and trance intensity is reciprocal inhibition. DMN and prefrontal areas help maintain Ordinary States of Consciousness (OSCs) and their inhibition correlates with a wide variety of ASCs, including shamanic trance. Similar “inhibition of inhibition” process was observed in psilocybin-induced OSCs (Carhart-Harris et al, 2012) and in Dietrich’s (2003) functional hypofrontality studies.

Thank you for directing our attention to this research, we have added it to our text.

Discussion:

  1. Lines 217-220: The sentence “Of particular interest for neuroscience is the correlation of the associated experience or perception of the altered state with brain imaging data with our work the first examining perceptual block task related fMRI thus we have greater spatial resolution than.” – needs to be revised

Thanks for noticing this error in the text – has now been corrected.

  1. Lines 222-225: The significance of these references is unclear as no specific findings are discussed. In addition, reference 16 is not available on the website provided (it refers to hypnotic rather than shamanic trance). Suggest omitting or discussing further and providing appropriate links.

These have now been omitted.

  1. Lines 246-247: “…we took a different approach by asking the subject when trance perception was high and low and relied on behavior as a model to run our analysis.” It remains unclear what behavior was relied on – please specify.

The behaviour is visualisation of the trance state, corroborated by subjective experience. This has been clarified in the text.

  1. Lines 261-265. Blanke’s research needs to be referenced directly other than in one of the author’s thesis (ref. 17) if the relevant brain region data is to be used for discussion.

Direct reference for Blanke has been added and this section has been improved.

  1. Line 267: Are the authors able to provide more specific localization than a general “right parietal region”? One useful reference source to correlate would be Michael Persinger’s data from Laurentian University (2010).

A reference for Persinger’s work has been added to this section.

  1. Lines 291-292: “we did not use visual stimuli to functionally activate area prostriata, but rather music associated with the subject’s ongoing 292 trance processes.” It would be important to specify in the Methods section whether the subject was asked to keep her eyes open or closed during the scan.

The subject was given the choice of eyes open or closed; she reported starting with eyes open then closing them early in the scanning process due to scanner noise etc. We’ve added these details to the text.

Conclusion:

  1. A statement on the limitations of this study should be made.

A statement of limitations has been added to the conclusion (Line 340 - 342).

References: 

  1. Northoff, G & Heinzel, A. (2006). First-person neuroscience: A new methodological approach for linking mental and neuronal states. Philosophy, Ethics, and Humanities in Medicine, 1:3
  2. Shapiro, Y & Scott, JR (2018). Dynamical Systems Therapy (DST): Complex Adaptive Systems in psychiatry and psychotherapy. Chapter in: Handbook of Research Methods in Complexity Science: Theory & Application. Prof. E. Mitleton-Kelly, Prof. A. Paraskevas, C. Day (Eds), London: Edward Elgar Publishing LTD
  3. Carhart-Harris, R. L., Erritzoe, D., Williams, T., Stone, J. M., Reed,L. J., Colasanti, A., … Hobden, P. (2012). Neural correlates of the psychedelic state as determined by fMRI studies with psilocybin. Proceedings of the National Academy of Sciences, 109, 2138–2143.
  4. Dietrich A. (2003). Functional neuroanatomy of altered states of consciousness: the transient hypofrontality hypothesis. Consciousness and Cognition, 12(2): 231–256.
  5. Persinger, M. A., Saroka, K. S., Koren, S. A. & St-Pierre, L.S. (2010). The electromagnetic induction of Mystical and Altered States within the Laboratory. Journal of Consciousness Exploration & Research, 1(7): 808-830

These have been added, along with

Neher A. Auditory driving observed with scalp electrodes in normal subjects. Electroencephalogr Clin Neurophysiol. 1961;13:449-51.

Newberg, A., D’Aquili, E., Rause, V. Why God Won’t Go Away: Brain Science and the Biology of Belief. Ballatine Books.

Newberg, A., Wintering, N., Waldman, M.R., Amen, D., Khalsa, D.S., Alavi, A. (2010). Cerebral blood flow differences between long-term meditators and non-meditators. Consciousness and Cognition, In Press.

Reviewer 2 Report

I do appreciate the originality of the aim focused in the study. It is a single case study, but it may be the basis for future study design.

Author Response

Thank you for your review of our work.  We've extensively edited our manuscript and had it checked for English by an editor. 

This manuscript is a resubmission of an earlier submission. The following is a list of the peer review reports and author responses from that submission.